# Faith as a Mechanism for Health Promotion among Rural African American Prostate Cancer Survivors: A Qualitative Examination

**DOI:** 10.3390/ijerph18063134

**Published:** 2021-03-18

**Authors:** Raymond D. Adams, Waldo E. Johnson

**Affiliations:** 1Department of Social Work, Psychology & Counseling, College of Education, Humanities, and Behavioral Sciences, Alabama A&M University, Normal, AL 35810 1, USA; 2Crown Family School of Social Work, Policy, and Practice, The University of Chicago, Chicago, IL 60637 2, USA

**Keywords:** psychosocial well-being, rurality, qualitative study, social work, African Americans, counterstories

## Abstract

Conceptualized using critical race theory as a theoretical underpinning, this study analyzed the lived experiences of older, rural, African American male prostate cancer (hereafter referenced as PrCA) survivors’ faith and health promotion practices within Northeast Louisiana. Qualitative data from journaling, observations, and semi-structured interviews were obtained from ten older, African American male PrCA survivors residing in four rural parishes of Louisiana. The data analysis employed a two-stage approach known as Polkinghorne’s analysis of narratives and narrative analysis using an art-based methodological approach. Framed as composite character counterstories, survivors’ narratives revealed how survivors made sense of and gave meaning to their PrCA diagnosis, treatment, recovery, and survivorship. Specifically, their counterstories indicate that centering and honoring the unique and often taken-for-granted perspectives of older, rural, African American male PrCA survivors offered a deeper understanding of the multiple factors influencing their quality of life, as well as the sociostructural mechanisms impacting their survivorship care. Faith was examined as both a secular and sacred source of support that these men viewed as central to the acceptance of their diagnosis, treatment, recovery, and survivorship.

## 1. Introduction

### 1.1. Prostate Cancer Survivorship and Health Promotion

Prostate cancer (PrCA) is the most commonly diagnosed cancer in men and is the second leading cause of cancer deaths among men in the United States, following lung cancer [1]. Most of the health literature has focused heavily on prevention, treatment, and management by specifically addressing patients’ diet, exercise regimen, lifestyle changes, and weight management [2,3,4,5,6]). Until recently, few empirical studies have explored the psychosocial functioning of African American PrCA survivorship, particularly among those existing in rural areas [7]. In sharp contrast, rural regions of the nation are routinely viewed as having limited access to advances in health care and treatment, as well as varied options for medical and health care [8]. Advanced directives for screening and early detection notwithstanding, hegemonic masculinity and medical distrust, which are prominent among African American men, could contribute to their higher rates of death from PrCA [8,9]. In addition, research has shown that even among those who survive PrCA, African American rural adults experience worse survivorship outcomes, including erectile dysfunction and enuresis [10,11,12,13,14]. Though no direct cause has been identified for these disparities, it is speculated that mental health status and services access, social networks, and spiritual factors are robust correlates [11,14,15,16]. 

Since the 1990s, prostate cancer deaths have dropped for all men, most of all for African American men. Yet, African American men have a greater risk of a PrCA diagnosis and a more aggressive type than white men. It takes a grim toll; African American men are twice as likely to die from the disease. Access to care, genetics, environment, and lifestyle are among the factors that most impinge on PrCA survivor rates among African American men [17]. This study examines PrCA survivorship among older African American men residing in rural northeast Louisiana. The authors documented their use and reliance on faith as a social and spiritual component of their decision-making process to seek detection and medical treatment, the selection of their respective treatment regimens, and survivorship strategies.

While some research has been conducted on the correlates of PrCA mortality between Whites and African American men [18]. far less work has been conducted specifically and exclusively on groups of PrCA survivors with African ancestry, especially those in rural areas ([19,20,21]. A case in point, cancer survivorship is understudied within the broader research literature even though “There are 14.5 million cancer survivors living in the United States, with most commonly diagnosed cancers of the breast (41%) in women and PrCA (43%) in men” [22] (p. 2). Equally important, a study conducted by Dickey and Ogunsanya (2018) found “that results indicate the need for additional studies that examine the factors impacting the quality of life (QoL) among Black PrCA survivors, using a theoretical framework so as to develop culturally appropriate interventions for Black PrCA survivors” [23] (p. 1648).

### 1.2. African Americans’ Use of Religion as a Source of Spiritual and Secular Faith

Although a considerable amount of published literature explores rural and urban differences among cancer patients and survivors ([24,25,26,27], few have examined the nuanced differences in their social connectedness to their community and faith. Within these articles are contrasting findings regarding disparities that are unique to rural and urban cancer survivors in different geographic regions. Not included within the larger body of literature is how rural cancer survivors organize their lives around a set of social determinants of health (i.e., economics, education, structural, and environmental racism) in contrast to their urban counterparts [24,28]. For this reason, this article purposefully centers the lived experiences of older, rural, African American male PrCA survivors by theorizing their survivorship care through a different methodological and conceptual lens. As revealed by Baldwin et al. (2013), “Rural cancer survivors are at a greater risk for a variety of poor health outcomes, even many years after their cancer diagnosis, and should be a target for interventions to improve their health and well-being” [24] (p. 1050).

Interventions that too often go understudied among rural and urban PrCA survivors are the impacts on their mental health status and access following their diagnosis [29]. As can be seen in the studies presented to date, more research is needed that gives a deeper understanding of how social determinants of health and well-being affect African American PrCA survivors [23,30,31]. This study explored the study participants’ engagement and reliance on faith as a mental health support mechanism and religion as “an important resource of resiliency for many racial and ethnic populations” [32] (p. 1). Nguyen (2020) further notes that given religiosity’s sociocultural salience among African American and Latino lives throughout their lifespan, it becomes paramount to situate mental health and well-being as a part of their aging process.

Religion is a major force in the lives of many African Americans. It is an important resource for African American individuals and families [33]. According to the Pew Research Center (2015), 79% of African Americans identify as Christian and an additional 3% identify with other non-Christian faiths. The African American population in the United States is predominantly Protestant, with close to 49% of blacks identifying as Baptist [34]. In Black communities, religion and the church serve many functions above and beyond spiritual sustenance [35]. The Black church is a social, cultural, civic, educational, and political institution in addition to being a religious institution that is central to Black communities [36]. Because of social, economic, and institutional disenfranchisement, African Americans have historically experienced difficulty accessing public and private services. As a result, Black churches tend to offer a greater number of community programs and mental health services than White churches [35,37]. Nguyen et al. (2019) report that, as a group, Black Americans have relatively high rates of religious involvement. [34]. Studies show that religiosity increases with age. Older Blacks attend religious services more frequently than younger Blacks, and compared to younger Blacks, older Blacks are less likely to never attend religious services. In fact, more than half (52%) of older Blacks, compared to 38% of younger Blacks, attend religious services either nearly every day or at least once a week [38]. Subjective religiosity also varies by age: older Blacks are more likely to attribute greater importance to religion in their lives and have a stronger religious identity than younger Blacks [39]. Finally, the use of clergy to address serious personal problems increases with age [40]. Taken together, research consistently demonstrates that older Blacks have higher levels of religiosity across a number of dimensions than their younger counterparts.

### 1.3. African Americans and the Spiritual and Secular Faith Tradition

Within the Judeo-Christian tradition, faith is recognized as a strong belief in God or the doctrines of a religion, based on spiritual apprehension rather than proof. Hebrews 11:1 defines faith as the substance of things hoped for, the evidence of things not seen. Adams (2019) employs a composite character counterstorytelling method in order to parallel the study subjects’ psychosocial life journeys through PrCA diagnosis, acceptance, treatment, recovery, and survivorship with their spiritual journeys as strong accomplished Black men who are afflicted with a potentially lethal cancerous disease [41]. Among the psychosocial and spiritual supports that the men acknowledged is faith. Moss (2020) affirmed that “within Black spirituality, all aspects of living are sacred. There is no separation of sacred and secular: singing, caring for your children, working and jogging, … the idea of caring for your body is also a sacred act” [42] (p. 2). Moss’ prophetic articulation of the Black cultural tradition underscores the origins and rationale for the merging of the sacred and the secular.

The melding of the blues and gospel traditions is a cultural phenomenon that is common among residents in the Mississippi, Louisiana, and Arkansas regions of the USA, especially older African American residents. This is, in part, a recognition of the intertwined relationship between the two musical genres and the fluidity of emotions and life course experiences of African Americans residing in the region and beyond in the diaspora. The blues is recognized primarily as an American music genre and is also widely credited as an African American contribution to American culture.

Yet, the blues tradition origins are within the Black gospel music and Black church traditions, which is further affirmed by the fact that many of the nation’s great rhythm and blues artists’ humble beginnings can be traced back to gospel music, including B.B. King and Sam Cooke, as well as some who began as blues artists and subsequently converted to become gospel artists, including Pop Staples. There is also the historical recognition that many secular musicians, including blues artists, supplemented their earnings by performing as church musicians. As such, many musical artists toggle back and forth, such as the celebrated Mavis Staples, so too does Black culture in its embrace of both the spiritual and secular expressions of culture being largely inseparable as experiential in Black life [42]. The interplay between these music genres, blues and gospel, is akin to the interplay and interdependence upon secular and spiritual faith that characterize the lived experiences of the study participants. The blues and gospel influences are widespread in the communities and are widely regarded and embraced by residents, especially those of the participants’ generation. They are not unlike other African American male and female peers who share and depend upon both secular and spiritual sources of faith that are integral to their daily lives [42].

Utilizing Northeast Louisiana as the geographical location of this study enabled the authors to draw upon the surrounding Mississippi and Arkansas areas, in addition to other areas of Louisiana that are largely rural but widely recognized as the American landscapes that birthed the blues and gospel music tradition. Many of the artists referenced above, as well as others in these other rural and urban areas of these Southern states, are native sons and daughters. Other American musical forms, including jazz, as “cultural cousins” are also among these areas’ contributions to American culture [43].

Faith as exhibited among these men appeared evident, for example, in their religious and spiritual relationships. The study findings suggested that the men drew spiritual and secular strength and emotional support from their fellow church members and the spiritual leaders. In contrast to the street affirmations of respect that Anderson (1994) references that many urban African American youth and young men often seek in their daily street encounters and one-off institutional interactions, “decent” or older African American men (or “old heads” as Anderson refers to them) sought respect and support via membership and leadership in institutions [44]. They sought membership in groups such as religious, civic and social organizations and mutual aid societies, including the Masons, Shriners, and Black Greek Letter Organizations (BGLOs) but also referenced as Divine 9) when such affiliation opportunities in the American social structure were inaccessible due to racial and gender discrimination [45].

For African American men, placing faith in individuals, such as pastors and clergy, or even institutions, such as churches and religious denominations, is not easily anchored. Far too many of these African American men are skeptical of sharing their reservations and fears with other men for fear of being duped [9] and have often operated outside of structural and social institutions due, in part, to racism, gender bias, and exclusion. There are instances in which the church and religious beliefs and practices parallel secular misnomers that discouraged help-seeking behaviors among men that are grounded in hegemonic masculinity and have often resulted in similar distrust issues for men. For example, Pitts (2018) argues that there remain strong beliefs in the Black church that mental health illnesses, such as depression and anxiety, are the results of the lack of faith or can be “prayed away” [46]. Various forms/tropes of hegemonic masculinity encourage self-reliance that minimizes reaching out for help. Although the sources of these guideposts are different, they often result in similar outcomes for the adherents.

Wilcox and Wolfinger (2016) suggest that faith, or the structured activity in which men engage in the Black church, may help them flourish in America, as its structure and values instill and strengthen employment prospects and family life [47]. The men in this study resemble a sizable swarth of Black men who have struggled with seemingly insurmountable psychosocial and health challenges. They often need support that offers a “magnificent mosaic of the human condition,” that is, a “blues note gospel that preaches about tragedy without going into despair” [43]. The men in the study have faced ominous times in their lives, including the PrCA cancer diagnosis shared among each of them, where their survivorship reflected this shared blues note gospel experience.

Secular faith is almost synonymous with trust. That is, one has to believe in the reliability, truth, ability, or strength of a person or things. The PrCA survivors not only placed faith in religious beliefs and individuals but also family and loved ones. These individuals offered advice and counsel, physical and emotional support, and care, among other investments, during the men’s diagnoses, deliberations, treatments, recoveries, and now survivorships. Some of these individuals sustained their support from the diagnosis through survivorship while others who provided support entered and exited their lives over time and throughout their PrCA journeys.

### 1.4. Framing and Contextualizing Rurality

Operationalizing the term rural [48,49] for this study posed a significant hurdle. In the field of social sciences, various definitions of rural are employed. This term has often overlapping, even slightly confusing meanings. Thus, the authors remained sensitive to the ways how the labels rural vs. urban/urban vs. suburban can evoke a myriad of sociohistorical, economical, and political conceptualizations and interpretations [50,51,52]. Nevertheless, scholars such as Hardy, Wyche, and Veinot (2019) recognized that “rural research findings have primarily addressed infrastructure and distance/geographic isolation as unique rural characteristics” (p. 1). Similarly, they hypothesized rurality through three separate approaches: (1) descriptive rurals, (2) sociocultural rurals, and (3) symbolic rurals, which, if synchronized, can help to establish more deep and nuanced interpretations of rural areas; in addition, they can aid in ways to “understand the relationship between technology and place” [53] (p. 3). To fully understand rurality, Hardy, Wyche, and Veinot (2019) found it necessary to define each of the three approaches. First, descriptive rurals generally place an emphasis on descriptive sociogeographical factors (e.g., population size, density, economics) to constitute an area as rural (Hardy, Wyche, and Veinot, 2019, p. 3) [53]. 

This definition reflects how data is utilized by state and federal organizations to showcase population density and socioeconomic levels. For example, the U.S. Census Bureau (2019) reports that African Americans account for 32.7% of Louisiana’s overall population [54]. Equally important is the fact that “one in three Black Louisianans lived below the federal poverty line in 2015 and were almost two and a half times as likely as whites to live in poverty” (Louisiana Budget Project, 2016) [55]. Second, sociocultural rurals are places that “highlight the extent to which people’s sociocultural characteristics vary with the type of environment in which they live, such as community values and perceptions of rural culture” (Hardy, Wyche, and Veinot, 2019, p. 5) [53]. This definition is relevant given the scope of this study. In particular, a study conducted by Griffith (2007) found that “People who reside in rural areas tend to be older and sicker than those who reside in urban areas, and they also tend to be poorer, which is often a function of environmental features, access-related factors, and community characteristics” [20] (p. 75). Third, symbolic rurals can be designated as areas that “focus upon how people use words, concepts, and feelings to connect themselves to the rural” (p. 5). This definition captures a number of important features of a rural area; chief among them is how
The various symbols and the meanings attributed to rural areas by inhabitants and outsiders allow for the “the rural” to manifest as something that can be understood at a scale beyond the local. This contributes to perceived socio-cultural differences between urban and rural areas described above.(Hardy, Wyche, and Veinot, 2019, p. 5) [53]

These approaches encapsulate a variation in how to articulate the concept of rurality while simultaneously maintaining a perspective that is manifested within their context. In the section that follows, the authors describe the unique features that are inherent within African American men’s social ecologies (e.g., church groups, churches, spouses, faith-based affiliations, barbershops, fraternal groups, sports).

As explained earlier, the authors provide clarity in how rural location or rurality impacts the psychosocial functioning of older, rural African American male PrCA survivors; in particular, those who exist within the contexts of the aforementioned approaches. As such, the authors extend previous research examining the racialized and gendered experiences of older, rural African American male PrCA survivors by using Cooks’ (2013) “composite characters” to frame their exploration within the context of the theory of critical race (Delgado and Stefancic, 2012) [56]. 

## 2. Materials and Methods

### 2.1. Research Questions

The characteristics of faith, as understood by older, rural African American male PrCA survivors, have undergone little discussion. Drawing from data (*n* = 10) previously collected through November 2018–May 2019, the authors posed the following two research questions: (1) How do older, rural African American male PrCA survivors view faith’s impact on their physical and psychosocial well-being? (2) In what ways do psychosocial factors influence the quality of life among older, rural African American male PrCA survivors?

### 2.2. Data Collection and Recruitment

Table 1 shows the participant profiles of ten older, African American male PrCA survivors from rural Northeast Louisiana.

## 3. Results

There is increasing knowledge among social and health scientists that recruiting African American men for social research and clinical trials, including PrCA research, must be done strategically [57,58]. In the dissertation study from which the present study sample is drawn, four recruitment approaches were employed as “successful recruitment strategies” (p. 2) to recruit older rural African American male PrCA survivors: (1) men who were aged 50 or older, (2) having a localized PrCA cancer diagnosis (stages I–IV), (3) at least six months postdiagnosis, and (4) residing in one of the 35 rural parishes in Louisiana [59].

The age range among the PrCA survivors at diagnosis in the study sample was 65–78. Several of the PrCA survivors shared information about family histories of PrCA specifically, while others referenced family histories of other forms of cancer, including breast cancer. The study participants were selected from a range of PrCA treatment options based on their stage of cancer at diagnosis. Early-stage diagnoses (I and II) yielded more available options for treatment.

### 3.1. Analysis

#### Composite Characters Storied Through Critical Race Theory

This present study employed the secondary analysis [60] method of Adams’ (2019) dissertation study, which explored the nexus between mental health, social networks, and spirituality among older, rural African American male PrCA survivors [41]. Recognizing that “narratives and storytelling may be particularly effective in soliciting personal and health information from minority populations and racial/ethnic groups with a rich tradition of storytelling” [61] (p. 2), the authors categorized the participants’ narratives through Cook’s (2013) critical race theory (CRT) method of composite counterstorytelling, which has three specific aims: (1) “provide the empirical space for researchers to recount the stories and experiences of people in politically vulnerable positions”; (2) serve as a “vehicle to present counter-stories that necessarily require descriptions of rich, robust contexts in which to understand those stories and lived experiences while maintaining the complexity of the meaning”; (3) “appeal to making research accessible beyond academic audiences” [62] (p. 182).

These aims, according to [63], allowed for the necessary space for the authors to disseminate the varied and substantive stories of participants via different mediums (e.g., biographies, interviews, journals, etc.) without academic analytic restrictions (e.g., hierarchal positivism). Conceptualizing faith within the critical race theory tenet of composite character counterstories for older, rural African American males PrCA survivors elucidates how health research has inadvertently silenced their shared stories through quantitative methodologies [64].

By reporting the findings through a CRT counterstory, the readers are given a glimpse into how these older, rural, African American male PrCA survivors made sense of and gave meaning to their survivorship through their faith traditions. More importantly, by presenting the stories as a Black aesthetic of their religious or spiritual identity, it “teaches others that by combining elements from both the story and the current reality, one can construct another world that is richer than either the story or the reality alone” [65] (p. 36).

### 3.2. Arts-Based Narrative Inquiry

Initially, ten older, African American male PrCA survivors from rural Northeast Louisiana participated in the study after a purposeful snowball sampling approach. Each participant engaged in two in-depth, semi-structured interviews regarding their experience with PrCA survivorship through Polkinghorne’s (1995) narrative analysis and analysis of narrative [66]. Through this analysis, the authors centered on each participant’s unique processes, linguistics, and cognitions by employing Polkinghorne’s (1995) two-stage narrative analysis [66]. Leaning into Collins’ (1986) concept of “outsider-within,” the authors recognized the participants’ shared perspectives (Polkinghorne, 2007) as African American male PrCA survivors was important to their overall identity [67,68]. The authors’ usage of Polkinghorne’s (1995) analysis of narrative, which seeks to identify themes across storied experiences, in conjunction with an arts-informed narrative inquiry approach [66,69] aims to not only “ensure the credibility of findings in relation to qualitative research” [70] (p. 1) as it pertains to this secondary qualitative analysis but also nuance the participants’ narrations.

In our collaborative attempt to remain close to the data, both authors engaged in applying Berger’s (2015) strategy of reflexivity and Saldaña’s (2015) method of member checking [71,72]. This engagement ensured participants’ experiences expressed their full meaning thematically and in alignment with the theoretical tenets of critical race theory. Our sensitivity to reanalyzing qualitative data [73] became a crucial element. As collaborators, we took into deep consideration the philosophical and cultural contexts of how the participants’ “culture influences health behaviors and the meaning of illness” [74] (p. 1).

Thus, the purpose of applying Polkinghorne’s (1995) analysis of narrative and narrative analysis is twofold [66]. In the first stage, in “narrative analysis or narrative mode of analysis” (Kim, 2015), the primary goal is “to help the reader understand why and how things happened the way they did, and why and how our participants acted in the way they did” [75] (p. 197). In the second stage, the analysis of the narrative, or as Kim (2015) denotes, the “paradigmatic mode of analysis,” the authors attempted to construct themes based on the survivors’ narratives with the clear intention of maintaining the richness of each story [75]. A cathartic restorying of faith was the primary aim of the authors rather than displaying the full usage of the analytic range of Cook’s (2013) CRT method of composite counterstorytelling. In short, the authors did not purposefully choose to omit certain “narratives” out of the stories [64].

The authors agreed that when providing counterstories, a sacred relationship with participants where not all of their stories reached the threshold of public consumption was to be maintained. This was not only for the protection of the participants by not revealing details that could bear the risk of potentially being traced back to the participants by people who might know some details of the narratives but also by using Cook’s (2013) CRT method of composite counterstorytelling [64], we were compelled to resist the voyeuristic addiction of cultural outsiders (e.g., Whites) about Black experiences. Therefore, the authors made the decision of what might be the safest ways to present counterstories without bringing harm to the participants. It is vital that the readers of these counterstories understand that our loyalty lay first with the participants and then with the alignment of our CRT-informed theoretical, ontoepistemic framework [64,76,77].

### 3.3. Findings

In this study, faith was conceptualized within the Black cultural tradition in which spiritual faith, namely, a strong belief in religion or doctrines of a religion based on spiritual apprehension rather than proof, was married with secular faith, namely, belief in the reliability, truth ability, or strength of a person or things. As exhibited among the African American male PrCA survivors, faith appeared evident, for example, in their spiritual and secular relationships. Table 1 provides information about the participants’ faith networks, both spiritual and secular. Each participant was allowed to choose spiritual figures and sacred text(s) that best embodied their sense of identity and experiences along their respective PrCA survivorship journey. As ascertained from Table 1, faith served as a powerful mechanism by which the participants could use to offset the multiplicity of institutional and structural barriers encountered during the help-seeking process. Although six themes were elucidated in the original study, the authors found that faith centered the embodied perspectivism of each participant.

### 3.4. Family and Friend Networks as Supports and Resources for Access to Care

Several of the PrCA survivors in the study reported a history of PrCA in their families, including Martin Luther King Jr. (MLK) and David. Matthew’s father was diagnosed at stage 4 with PrCA and subsequently died. Other PrCA survivors in the study report having other chronic health challenges or having been survivors of other potentially lethal health issues. John the Disciple reported that he suffered from coronary heart disease and recovered from a heart attack. Paul the Apostle reported having multiple comorbid disorders, including diabetes, cataracts, hypertension, and peripheral neuropathy.

Prior family experiences with PrCA may deepen knowledge of the disease’s etiology and broaden PrCA patients’ perspectives to resist viewing cancer through a fatalistic framework when selecting treatment options and viewing their patient status in terms of survivorship [1]. Engaging in frank and honest discussions about PrCA with family, friends, and fellow members of their church and club/organization communities promoted faith and survival strategies. Female members of their support networks who had personally experienced cancer or via other female family members and friends may have served a uniquely supportive role in the PrCA survivorships. Ravenell, Whitaker, and Johnson (2008) found that in generating focus group discussion with Black men about colorectal cancer detection, the male participants were routinely uncomfortable about undergoing colorectal cancer examination or admitting they had been screened for colorectal cancer [9]. As a result, generating discussion about their examination and detection experiences was challenging. However, when asked about female members in their families who had been screened for breast and other forms of cancer, the tense atmosphere shifted and the conversation triggered a readiness to explore gender differences in types of cancer, as well as cancer detection among men [9].

Jones et al. (2008) found that prostate cancer affects not only the man with the disease but also his entire family [1]. Family relationships play an important part in how men with prostate cancer cope with the disease and decide on treatment issues. In the study, seven of the PrCA survivors reported that female family members played integral roles in their decisions to seek medical assistance to be examined for cancer detection, seeking a second opinion, treatment options, and other patient and family issues. Job 1:2’s wife accompanied him to the urologist and was diagnosed in her presence. Matthew credits his ex-wife, a physician, for her positive contributions toward his PrCA survivorship. David’s wife, a retired registered nurse, encouraged him to seek a second opinion and supported his decision to seek treatment at MD Anderson in Houston, which is an urban network of cancer treatment centers that are widely recognized for their advanced technology. Little James also credited his wife, a pediatrician, for recommending that he get a second opinion and consider a range of treatment options. MLK first confided with his wife when he received his PrCA diagnosis. Paul the Apostle’s wife also advised him to seek a second opinion and consider a range of treatment options available to him. In addition, his brother, who received medical and health training during his military service, also encouraged him to seek a second opinion and the full range of options available for treatment given his diagnosis. Job’s girlfriend accompanied him to his medical appointments subsequent to his diagnosis and throughout his treatment. John the Disciple referenced his sister, niece, and his niece’s friend (a cancer counselor) as sources of advice and support.

As referenced above, the family and friend networks of five PrCA survivors who advised the men to seek second opinions following their initial diagnosis were professionally trained medical or allied health workers. Paul the Apostle’s brother, who worked in allied health during his tour of military service, advised him to seek a second opinion and a range of treatment options. David’s wife (retired registered nurse), Little James’ wife (pediatrician), Matthew’s ex-wife (physician), ad John the Disciple’s niece’s friend (cancer counselor) all contributed important professional information and advice that enhanced the quality of these men’s treatment and care decisions by recommending them to get a second opinion and to consider a range of treatment options. These men’s access to a network of individuals with education, training, and professional experience in healthcare provided the PrCA survivors with a level of support and enhanced their faith in ways that were unmatched by the faith they placed in other family, friends, church, and community members within their support network. This information network not only enabled these PrCA survivors to benefit from access to a medical professional network that is often inaccessible to many African American medical patients but also mitigated against concerns about the limited medical science and technology advances that residing in a rural area routinely presents. Collectively, these medical and non-medical supports strengthened the men’s faith in recovery and survivorship [9]. Family members formed the core of the PrCA survivors’ support networks. In addition to their wives and partners, their siblings, children, and other extended family members were identified as individuals in whom the PrCA survivors placed their faith to provide advice and support.

MLK identified his daughters, brothers, and his mother as his family sources of support in response to his diagnosis. John the Disciple reported that his sister and nieces formed his family support. Paul the Apostle referenced his brother as someone in whom he placed his faith to provide advice and resources.

### 3.5. Professional/Vocational Status as a Source of Faith and Access to Quality Care

An interesting artifact that was shared among many of the study participants is their educational attainment. Nine of the ten PrCA survivors in the study held college baccalaureate degrees and taught for decades in the area school systems, several of whom held graduate and professional degrees, including in divinity, law, and social work. The postal worker among the group held a college degree from one of the area’s historically Black colleges and universities (HBCUs). The one PrCA survivor who did not hold a college degree held three certifications, including alcohol and drug counseling (CADC) and public speaking. The snowball sampling approach to the data collection employed in this study’s methodology will likely yield participants with similar demographic characteristics. The fact that 9 of the 10 men in the sample were college graduates notwithstanding, these men faced challenges in navigating the healthcare system, making informed decisions about treatment options, and maintaining positive mental health while adjusting to a cancer diagnosis. Such experiences are less likely to have plagued their racially White peers.

Collectively, the educational attainment among these men positioned them in close association and affiliation with other community and area professionals, particularly healthcare professionals who could provide valuable information and advice that enhanced their decision-making regarding treatment options and aftercare. Several of the PrCA survivors were ministers, but these men were also veterans and members of church congregations and social and civic organizations in addition to their family and friend networks.

The study findings suggested that the men drew strength and emotional support from their fellow church members as well as the spiritual leaders. Five of the PrCA survivors (MLK, Little James, Paul the Apostle, David, Job 1:21) were ministers and pastors of congregations in Arkansas and Louisiana. As ministers or faith leaders, an abiding trust in a higher being was central to their spiritual identities. They professed their belief and trust in a higher being and by doing so, inspired others, especially their congregants or spiritual followers, to develop and deepen their own faith. As pastors of religious congregations, these PrCA survivors were institution builders that often help their membership to understand the relationship between the physical and the spiritual. Health status and challenges, such as PrCA, can be understood as not solely intervened or treated from a medical treatment perspective but also treated with spiritual healing [43]. Within the Black church tradition, ministers and pastors routinely employ the use of personal stories in their sermons and teaching opportunities in which they affirm their faith in a spiritual power, along with faith in secular support and interventions, to overcome challenges and hardships. Frequently, they attest that faith alone delivered their conquests over such challenges and hardships when secular interventions appeared insufficient [36].

MLK reported that his two church congregations (he led congregations in Louisiana and Arkansas) and his wife (as the first lady of those congregations) were sources of spiritual faith when he shared news of his diagnosis. The congregations prayed “unceasingly” on his behalf for recovery and received help from church support groups. Little John shared his PrCA diagnosis with a church congregation member and together they performed a “cooperate prayer.” He found that to be an honoring and humbling experience. In contrast, Paul the Apostle did not share his PrCA diagnosis with his congregation. He stated that he shared his diagnosis with his support network but beyond his wife, it is unclear who composed that network. He did not offer an explanation for his decision not to disclose his diagnosis to his congregation. It is unclear whether Job 1:2 shared his diagnosis with his congregation, although he provided an extensive dialogue about his diagnosis, faith, and his calling (call to the ministry). Job 1:21 professed that his call to ministry was stimulated by his belief that as a minister, he could rehabilitate juvenile delinquents following 40 years of work as an educator and juvenile detention counselor. Beyond these corporate expressions of faith in which these men were engaged, as both leaders and recipients of faith works, they shared information about their personal spiritual faith.

In addition, Job 1:21, Paul the Apostle, and Job were U.S. military veterans, which is a status that afforded them access to material and instrumental services and institutional support post their respective service exits for the remainder of their lives and some possible support to their families. Job 1:21 also referenced, in contrast, having been exposed to Agent Orange as a Vietnam veteran, suggesting that his military service might have compromised his health. Benefits and drawbacks notwithstanding, participation in religious and other organizational and institutional affiliations may have provided interpersonal fellowship that enabled them to build confidence in their treatment strategies, as well as support in knowing that fellow participants in these organizational and institutional settings are encouraging their efforts. Ravenell, Johnson, and Whitaker (2006) found that African American men that were fifty years and older perceived better health outcomes resulting from their participation in organizational and institutional fellowship than younger Black men [44,78]. Here we reiterate that Anderson theorized that “decent” or “old heads” (as Anderson refers to older urban Black men) seek respect via participation and leadership in religious, civic, social organizations, and mutual aid societies, including the Masons and Shriners, and even BGLOs (which are also referenced to as Divine Nine or D9) when, historically, such affiliation opportunities in the traditional American social structure were inaccessible due to racial and gender discrimination [45].

### 3.6. PrCA Survivors’ Perceptions about Interactions with Medical and Healthcare Providers

Faith in the family provision of support is critical because Black Americans have a broad distrust in the medical and healthcare system. This is currently reflected in the lower percentage of African Americans who are prepared to take the vaccination for the coronavirus disease 2019 (COVID-19) virus. Forty-two percent of African Americans, compared to 83% of English-speaking Asian Americans, 63% of white Americans, and 61% of Latinx Americans, expressed willingness to get the COVID vaccination [79]. Much of this reluctance and medical distrust is grounded in historic examples in which African Americans have been duped by public health officials in studies, such as the Tuskegee Experiment of the 1930s, when researchers lied to hundreds of Black men, telling them they were conducting research on treatments for “bad blood.” In reality, the scientists were allowing Black men to die of untreated syphilis [79]. The experiment was slated to go on for six months but lasted for 40 years. African American men are especially distrustful of medical and public health initiatives [80]. Little James expressed concerns about medical mistrust and belief that cancer is a pathogen that presents as a virus and manifests as cancer. Although his understanding of the etiology and development of cancer is basically accurate, misinformation and conspiracy theories also contribute to African Americans’ distrust in the medical and healthcare fields, often emerging out of past racial injustices and shared personal experiences when seeking medical and health interventions [79].

In addition, gender differences in the orientation of male and female health engagement possibly explain some of the differentials in willingness to engage the medical and healthcare system. Across race and class, female and male patients in the American healthcare system experience it differently [78]. Female patients are frequently introduced to a health engagement regimen that begins during childhood and continues thru adulthood in which they engage with a range of specialty medical and health providers. In contrast, male engagement with the healthcare system begins in parallel with female engagement but more frequently drops off during adolescence unless these youth are engaged in prep sports where physical examinations are required. Social class may explain some of the differentials, but in general, the continuity and coordination of healthcare among males over the life course may suffer. Sacks (2019) argued that Black middle-class women’s engagement with the American healthcare system falls far short of their White female social class peers, frequently resulting in adverse healthcare outcomes that parallel the experiences of poor African American women [81].

In the current study, several PrCA survivors offer insight into how their confidence and/or distrust in their medical providers affected their perspectives about PrCA treatment. MLK observed that when he received his PrCA diagnosis, there was no social worker available to assist him in processing his diagnosis and to begin to consider treatment options. He relied heavily on his family and church networks to fill this void and although he was able to do so, he lamented that this form of professional support should have been provided. The failure to provide informational and counseling support following a life-threatening medical diagnosis is viewed as a failure of the medical and healthcare system [81], and a potentially underlying contributor to the healthcare and medical distrust that is widely experienced among African American men [82]. MLK also opined that had his urologist been an African American health care provider, he might have been able to ease some of MLK’s hesitation about seeking treatment. The belief that having a healthcare provider who is race-matched and shares the life experiences of the patient enhances the patient’s medical and healthcare experience is well-documented [83,84]. In contrast, Joseph highlighted that his Filipino physician noticed signs that might indicate prostate cancer and recommended an oncologist to conduct further examinations. Paul the Apostle referenced his physician as an important source of support and displayed sustained faith in his physician’s aid. Joseph also displayed faith in his physician.

David and Little James shared information about their interactions with their primary physicians that suggested similar concerns. Upon receiving his PrCA diagnosis from his urologist, David chose to seek treatment at MD Anderson in Houston with a Black urologist. He was displeased by his urologist’s response when he informed his doctor that he was seeking a second opinion and treatment elsewhere. Little James, a lawyer and pastor of a small congregation, expressed concern that his doctor’s recommended options for treatment seemed not to consider impotency as a possible side effect and how that would affect his future plans to grow his family. He selected a radical treatment and 9 months later, he was concerned about incontinence and impotence. He stated “be logical yet skeptical” based on utilizing a combination of science and faith in decision-making. His exact words of advice to other men with a PrCA diagnosis were as follows:
Well, I would want them to know to be logical and skeptical. And if you see reasons to be skeptical, don’t let people, necessarily, make you feel that your skepticism is illogical, if it is in fact logical. Be careful of overoptimistic outcomes because most doctors try to find ways to paint their successes in the best possible light.

## 4. Discussion

The understudied aspects of prostate cancer (PrCA) survivorship among older, rural African American men that were highlighted in this qualitative study deserve further examination. This analysis offered a deep contextual understanding of how these men used their church affiliations, family, and social networks to overcome their personal fears and anxieties about a PrCA diagnosis and barriers to healthcare equality and equity that were largely based on gender, racial, and social class.

When faith is examined from a spiritual and secular perspective that is grounded within a larger Black cultural tradition, its value is increased multifold and exponentially enhanced. Even men within this study who professed no affiliation or involvement with any particular religious denomination or spiritual community shared perspectives and examples of how their immersion of the Black cultural faith tradition supported them throughout their bouts with cancer, from diagnosis to treatment and remission. Family and friend relationships offered similar, if not parallel or more generous, forms of instrumental, emotional, and knowledge-based support to those received from church and parish leadership and lay membership.

The ability of these men to garner and organize such extensive support networks is noteworthy. Social, medical, and public health science scholarship coincide in their depictions of men “going it alone,” particularly when dealing with health challenges [85]. That stoicism is perhaps even more widely attributed to African American men. The troubled history that characterizes African Americans’ experiences with the medical healthcare and public health systems illuminates the inherent distrust they share, despite their preponderance of chronic health problems [52,78]. African American men are distrustful of the medical healthcare and public health care systems. The historical mistreatment that African American men have endured in the medical and public health initiatives over time contributes to their wariness ([9,80]. 

In contrast, the participants in the study were surrounded by fellow church members and leadership, family, and friends throughout their cancer ordeal. They were also well-educated, which placed them in close proximity to social networks that could provide instrumental, emotional, and information resources [80]. As a result, their options for responding to their PrCA diagnosis were expanded beyond their geographical residences or their personal knowledge about treatment options.

The PrCA survivors acknowledged that improvements in the diagnosis to treatment process could be improved. They advocated for social workers to provide counseling to recently diagnosed PrCA patients. They also advocated for providing recently diagnosed PrCA patients with the full complement of treatment options.

## 5. Limitations

Our qualitative study’s findings are not generalizable. Our primary goal was to gather rich and substantive qualitative data [40,86] that spoke to the associative effects of mental health, social networks, and spirituality. The study also employed self-reporting and recall for elderly men after their PrCA treatment and recovery. While the men provided important and substantive information, there is the likelihood that the recalled data contained some inaccuracies. Concerns raised about the accuracy of self-reported data are appropriately raised here.

Using Polkinghorne’s (1995) two stages of narrative configuration as an analytic approach allowed us to explore the context in which participants made sense of their illness and faith [66]. According to Polkinghorne (2007), “Storied evidence is gathered not to determine if events actually happened but about the meaning experienced by people whether or not the events are accurately described” [68] (p. 479). This is another limitation of the study because the artistic nature of the narratives may detract from the authenticity.

The focus on older rural African American men and their survivorship experiences with PrCA is overdue and this study offered us important insight into their lives. However, the findings’ exclusive focus on rural men limits our understanding to a geographical subpopulation of African American elderly men with PrCA. Finally, this study focused on older African American men with PrCA. Although this population represents the majority of cancer survivors, there are young men who are also PrCA survivors. Their experiences are not represented in the stories shared by the men in this study.

## 6. Conclusions

This study has raised many questions in need of further investigation. As pointed out by Corley and Young (2018), “The daily lives of racial and ethnic minoritized groups continue to be affected by a racist system of hierarchy and inequity that characteristically advantages White Americans while creating detrimental outcomes for people of color” [87] (p. 318). With this in mind, the findings of this narrative study offer the following recommendations for future research.

Our findings suggest that meaning-making practices are key to nuancing interventions among older, rural African American male PrCA survivors. Each participants’ spiritual connectedness gave voice to how they understood, perceived, and experienced pre- and postdiagnosis issues. Future research should include a cross national rural study of older African American men. A natural progression of this scholarship could analyze the survivorship and meaning-making of inhabitants of the “Deep South” [88,89].

Using methodological analyses, larger datasets, and other research designs, these preliminary findings could be further investigated and expanded to explore other aspects of African American men and the healthcare system. Stewart et al. (2019) conducted a two-phase, mixed-methods study to understand how Black men think about primary care and usual sources of care [90]. Dean et al. (2018) state that “Despite the body of evidence supporting the importance of social factors to cancer, incorporating social factors such as patients’ race/ethnicity and socioeconomic position into cancer research and clinical practice continues to be a challenge in both risk assessment and survivorship research” [90] (p. 12). For these factors to have a substantive and transformative impact within the extant scholarly literature, a robust theoretical framework is necessary. Further research could use relevant theoretical frameworks to “Elicit survivors’ subjective meaning-making processes beyond the injury event” [91] (p. 1165).

## Figures and Tables

**Table 1 ijerph-18-03134-t001:** Participant profiles.

Participants	Age	Residence	Main Occupation	Sources of Faith	Stage of Cancer at Diagnosis
Little James Galatians 1:19	68	Lake Providence, LA	Lawyer	Wife was a pediatric physician/church members	II
Paul Romans 1:1	78	Rayville, LA	Minister/retired educator/veteran	Deceased wife/ church/grandchildren	I
John John 3:16	66	Monroe, LA	Unemployed	Family/church/niece/ nephews	I
Job Job 1:1	65	Monroe, LA	Postal worker/veteran	Girlfriend/ex-wife	I
Joseph Genesis 39:4	65	Shreveport, LA	Licensed clinical social worker	Wife/church	III
Martin Luther King Jr. Philippians 4:13	71	Monroe, LA	Pastor	Wife/church	IV
David Psalms 46:1–11	73	Monroe, LA	Minister	Wife (nurse)/church	II
Matthew Matthew 1:21	70	Monroe, LA	Veteran	Church/ex-wife/son	I
Job 3:16 John 1:2	66	Monroe, LA	Minister	Wife/church	I
Job 1:21 Roman 8:37	75	Monroe, LA	Minister/veteran	Wife/church	III

Note: Cook’s (2013) method of composite characters was used to maintain the confidentiality of each participant.

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
