# Peer review of "Faith as a Mechanism for Health Promotion among Rural African American Prostate Cancer Survivors: A Qualitative Examination"

_ijerph, 2021, doi:10.3390/ijerph18063134_

Round 1

Reviewer 1 Report

  1. I like the topic. Overall, you did a fairly good job. My main disappointment with the paper is that you failed to provide the Institutional Review Board (IRB) statement. This study involves humans, and we are ethically required to obtain an IRB approval for a subject of this nature. I teach research methods at the graduate level, among other things, and an IRB statement is one of the key areas I lay emphasis on for my students. Without an approved IRB statement from your institution, how are you able to prove the ethical probity of this research that involves humans? In our research, we are told to "Do no harm." How are we to accept that your research has done no harm when you violated the basic ethical and research procedure of going through the IRB for certification? On page 13, lines 514-519, you cited the ethical problem with the Tuskegee Experiment of the 1930s. If you know this, why did you not go through the process of the IRB of your institution before involving human subjects in your research? Though I like your paper, I cannot recommend it for publication until you provide the IRB statement required on page 16, lines 649-656.
  2.  The other major problem is that Table 1 which you provide on pp. 6 & 7, lines 272-273, is repeated exactly the same on pp. 9 & 10. How are the two different from each other? One would have thought that pp. 9-10 should be Table 2 with a different title from "Participant profiles" of Table 1, and that it (Table 2) would focus on the perceived effect of faith on their PrCA to prove that faith worked for all or for some or did not work at all (whatever conclusion reached by your research) as a mechanism for health promotion among the population you selected for study. But since the human subjects of your study are PrCA survivors, I think the foregone conclusion is that faith worked for all of them. That would then lead you to find out and tell us what/which faith worked for each person--religious faith or faith in the family/friends?
  3. Other things to take care of are related to thorough proofreading and editing: line 33--delete "s" from "lifestyles" since it is an adjectival noun qualifying "changes" on line 34; line 42--change "is" to "are"; line 84--delete "Our study...";  lines 89-92--indent the passage and give page reference; line 98--insert "(49%)" after "blacks" and delete "49%" before "Nguyen" on line 99; line 107--delete "s" from reports; lines 108-119--it seems you are quoting Nguyen et al. In that case indent the entire passage; line 115--"al" should be "al."; line 120--change "men-most" to "men--most"; line 143--delete quotation mark; line 145--change "Tradition" to "tradition"; line 164--change the first "are" to "as"; line 200--make it "blues-note gospel"; line 201--delete "belief" and replace with "one has to believe"; line 275--insert "for" after men; insert a comma after second "research"; line 276--delete "so"; line 286--insert "were" after "participants"; line 298--after "(2013)" insert "Critical Race Theory" before "(CRT)" since you are using it for the first time in a sentence; line 301--insert closing quotation mark after "positions;" and delete "s" from "serves"; line 304--delete "composite counterstorytelling is the" and replace the phrase with "..." to read "3) ....appeal"; line 310--delete "(hereafter CRT)"; line 325--delete "s" in "deviates"; lines 327-328--explain the difference between "narrative analysis and analysis of narrative" or adopt the less confusing style of Kim's "narrative analysis or narrative mode of analysis."; line 325--insert "by" after "also" and delete "of" after "using"; line 370--insert "heart" after "coronary"; line 376--remove period after "et" and place it after "al" before the comma; line 390--place a period after "al"; line 404--place a comma after "addition" and delete "to"; line 406--insert "to" after "him"; line 417--delete "for" and replace with "by"; delete "that he" and replace with "to them to"; line 418--delete "considering" and replace with "to consider"; line 419--delete "whose" and replace with "with"; line 476--after "and", insert "received help from"; line 480--after "that", insert "he shared"; line 486--change "a" after "as" to "an"; line 501--start the sentence with "Here, we reiterate that Anderson"; after "theorized" insert "that"; line 506 after "discrimination" insert "(Ross, 2000)."; line 520--insert a period after "...2019)"; line 528-- delete "s" from "explains"; line 531--delete "t" from "regiment"; line 550--delete "s" from "Americans"; line 599--incomplete sentence. what were you going to say?; lines 564-571--poor, weak sentences; reconstruct the sentences to bring out the sense, the meanings in them; line 581--delete "With that being said," and begin the sentence with "It has..."; lines 591-593--make a full sentence to make the meaning clear; line 631--hyphenate "mixed-methods" properly; line 636--make it "patient's".
  4. p. 16, lines 649-556--Institutional Review Board Statement. I have already addressed this at the beginning.
  5. line 681--fix this entry; page 18--rearrange the entries in strict alphabetical order where the C comes before D and where "Coats" comes before "Collins" and "Delgado" comes before "DeSantis"; p. 20--"Li" should come before "Lichter"; p. 21--"Murphy" should come after "Moss III"; p. 22--"Pitts" should come before "Polkinghorne"; line 903--change "RHIR" to "RHIH".                

Author Response

I am uploading two documents to address the reviewers' comments.

  1. IRB approval document: The authors have included a copy of the IRB approval document. It is located in the Unpublished Documents portal.
  2. Table I: We removed the duplicated Table 1 illustration from the document. There is now only one Table 1 in the document and it is located on page 7. With respect to your query about whether secular or spiritual faith worked for each PrCA survivor, the authors contend in the theoretical definitions of faith (pages 2-5) paper conceptualizes faith, as experienced among these men, as both secular and spiritual. The authors provide subsequent support throughout the findings (pages 9-13) and conclusion (page 15) sections of the manuscript that  to support was both secular and spiritual.
  3. Proofreading and editing: We have responded to each proofreading and editing call-out in the reviewer’s comments. The current manuscript draft includes those edits throughout the document.
  4. Revised Reference Pages: The authors have accurately alphabetized the reference pages (16-23).

Reviewer 2 Report

The article addresses the importance of faith among Rural African-American prostate cancer survivors in Louisiana. The introduction and the materials and methods are thorough and the findings are interesting. I have only two minor suggestions: (1) a reminder of the concept of faith in the Findings section or describe how faith has been categorised in the Findings--as we all have some preconceptions as to what 'faith' means; (2) the Discussion and Conclusion sections can be combined as they are both quite short and not entirely supported by the findings; (3) the limitations of the study can be shortened and incorporated in the Materials and Methods section.

Author Response

I am uploading the document to address the reviewers' comments. 

  1. Inclusion of the operationalized definition of faith in the Findings Section: The authors provide an operationalized definition of faith at the beginning of the Findings section on page 9. 
  2. Combining the discussion and conclusion sections of the paper: The authors revised the manuscript’s discussion section (page 14) so that it is supported by the study findings and is distinct from the manuscript’s conclusion (page 15).

Reviewer 3 Report

This is an important topic. The essay can be strengthened with attention to organization.

  • I am curious about the role of medical racism in the PrCA mortality rates amongst African American men.
  • Clarify some sentences such as this: 

    Our study ... The study data upon which this manuscript is 84 based does not employ standardized measures of mental health status but instead ex-85 plored the study participants’ engagement and reliance on faith as a mental health sup-86 port mechanism and religion as “an important resource of resiliency for many racial and 87 ethnic populations” (Nguyen, 2020).

  • This paragraph seems redundant and misplaced: 

    Since the 1990s, prostate cancer deaths have dropped for all men-most of all for Af-120 rican American men. Yet, African American men have a greater risk of a PrCA diagnosis 121 and a more aggressive type than white men. It takes a grim toll; African American men 122 are twice as likely to die from the disease. Access to care, genetics, environment and life-123 style are among the factors that most impinge on PrCA survivor rates among African 124 American men (Stuart, 2009). This study examines PrCA survivorship among older Af-125 rican American men residing in rural northeast Louisiana. The authors document their 126 use and reliance on faith as a social and spiritual component of their decision-making 127 process to seek detection and medical treatment, the selection of their respective treat-128 ment regimens and survivorship strategies. 

  • Can better transition into a discussion of the blues. There is definitely a connection, but it's not made clear here. The discussion of the blues seems paratactically connected to conversation of secular and sacred, but only for those familiar with both the blues tradition and Black church traditions.
  • Table 1 is inserted twice
  • This observation ignores studies that show that attainment of middle class status is often difficult and maintained precariously or not at all: 

    Although the snowball sampling approach to data collection employed in this 444 study’s methodology will likely yield participants with similar demographic character-445 istics, the geographical and socioeconomic dimensions of the communities sampled also 446 defy the likelihood that college educated older African American men would largely 447 comprise the study sample.

  •  

Author Response

I am uploading document which addresses the reviewers' comments

  1. The absence of a standardized measure of mental health status: The focus of the manuscript did not include a clinical assessment of mental health statuses of the study participants. The authors do not assess the mental health status of the study participants, nor do the authors verify the PrCA diagnosis or the treatment received by the study participants. The data analyzed by the authors are self-report sources of information provided by the study participants.
  2. The paragraph referenced as redundant and misplaced: The referenced paragraph has been edited so that any redundancy has been removed. The revised paragraph is now better situated within the manuscript. It is found on Page 2.
  3. Transitioning to the blues tradition discussion: A transition paragraph has been inserted to precede the blues tradition narrative referenced in the reviewer’s comments. It is found on page 3. The inserted paragraph transitions the discussion about the blues, the blues tradition and the African American church traditions to a discussion of an African American cultural tradition which blends elements of blues and gospel music in describing the lives of African American. The paragraph referenced in the reviewer’s comments focused on the Nguyen (2020) has been edited and is found on pages 2-3.
  4. Table 1: The second Table 1 has been removed from the document.
  5. The issue of middle class: The conceptual framing of the manuscript does not include a class analysis of the participants and therefore, does not focus on how neither their PrCA diagnosis nor recovery and survivorship impacts there class statuses. In describing the study participants, there is a reference to the educational attainment of the PrCA survivors solely as a characteristic of the sample. With the exception of one study participant, the remaining study participants are college graduates, several of whom hold advanced degrees. While education is regarded as a predictor of socioeconomic status in the US and one that is widely embraced by African Americans, the authors share the reviewer’s perspective that educational attainment is in a precarious predictor African Americans because of systemic racial bias, social bias and institutionalized racism. We concur with the point made by the reviewer that middle class status for African Americans is difficult to sustain. The authors made no attempt to link the study participants’ educational attainment to their individual or collective class status. Given that we used a snowball sampling approach for recruitment, the fact that we recruited a largely college-educated sample makes sense and is not questionable. As a result, the authors do not discuss the challenges among these men to sustain their class statuses nor do the authors provided previous studies to support this contention.

Round 2

Reviewer 1 Report

This is a much improved version. The IRB approval concern has been answered. The authors have improved the overall quality of the paper. They should now make the following minor corrections:

  1. line 301: delete closing quotation mark after Critical Race Theory
  2. line 304: change "serves" to "serve" 
  3. line 665: delete "study" after "African American men"
  4. lines 750-751: delete title underline and use italics
  5. 756-757: "Collins" entry should come after, not before, "Coats" (lines 759-761)
  6. line 815: delete title underline and italicize it
  7. Maintain consistency of order of publication year. Previous entries were from past to later/latest year, e.g. "Bhattacharya, K. (2019)" to "Bhattacharya, K. (2020)"; see lines 727-732. Also see "DeSantis...(2013)" and "DeSantis...(2014)", lines 778-782. Use that same style of year order for "Hill, A....", lines 817-821, so that 2018 comes before 2020, in order to maintain consistency
  8. line 831: delete title underline and use italics
  9. lines 834-835: enter the year of Johnson's unpublished manuscript; it could be the year it was written whether as a thesis, dissertation, conference, workshop, lecture, or forum paper       
  10. line 858: delete title underline and use italics
  11. 882-886: "Moss III...(2015). (2014)" should come before "Moss III...(2020)" to maintain consistency
  12. lines 910-915: "(Owens, O.L....2019a" should come before "Owens, O.L....2019b)"
  13. lines 935-940: "Ravenell, J. (2006)" should come before "Ravenell, J. (2008)
  14. lines 942-946: "Ross, L." should come before "Sacks, T."
  15. lines 977-987: "Taylor, R..." year order should be 2000, 2004, 2011, 2014
  16. line 983: italicize title and delete underline
  17. lines 995-1000: "Weaver, K.E. (2013a)" should come before "(2013b)" 

Author Response

Good Evening, 

Dear Drs. Thorpe and Bruce,

The authors thank you and the reviewers for detailed feedback you provide in the reviewers’ comments. We are happy to reply to those comments and believe that they indeed enhanced the quality of our manuscript. Below we provide a point-by-point reply to each reviewer comment and the location of the responses in the manuscript for easy review.

Reviewer #1:

  1. The authors have deleted closing quotation mark after Critical Race Theory it is denoted as being highlighted in yellow on line 301 on of the manuscript.
  2. The authors have changed “serves” to “serves” on line 304 it has been highlighted in yellow on of the manuscript.
  3. The authors have deleted “study” after “African-American men on line 665 it has been highlighted in yellow on of the manuscript on line 667.
  4. The authors have deleted title underline and use italics on lines 750-751.
  5. The authors have organized Collins (1986) citation located on line 764 after Coats (2017) citation on line 760 on.
  6. The authors have deleted title underline and use italics on lines 816 and is highlighted in yellow.
  7. The authors agreed with the reviewers in maintaining consistency therefore the authors have rearranged the Hill 2018 citation on line 822 before the Hill 2020 citation.
  8. The authors have deleted title underline and use italics on lines 831
  9. The authors agreed with the reviewer’s recommendation to include a year for the unpublished Johnson citation on line 839. It is highlighted in yellow.
  10. The authors have deleted title underline and use italics on lines 858.
  11. The authors agreed with the reviewers in maintaining consistency therefore the authors have rearranged the Moss 2015 2014 citation on line 877 before the Moss 2020 citation.  
  12. The authors have organized Ravenell (2006) citation located on line 941-942 and highlighted in yellow after Ravenell (2008) citation on line 944-946.  
  13. The authors have re-organized Ross (2000) citation located on line 948 and highlighted in yellow after Sacks (2019) citation on line 951.  
  14. The authors have re-organized the Taylor citations beginning on lines 983-992 and highlighted in yellow as Taylor 200, 2004, 2011, 2014.
  15. The authors have deleted title underline and use italics on lines 983.
  16. The authors have re-organized Weaver (2013a) citation located on lines 1002-1003 and highlighted in yellow after Weaver (2013a) citation on lines 1005-1006.
  17. The authors have re-organized Owens (2019a) citation located on lines 916-917 and highlighted in yellow after Owens (2019b) citation on lines 919-920.

Again, the authors thank the reviewers for their careful read and thoughtful review of the manuscript. The resulting revisions greatly enhanced the manuscript’s quality and contribution to the research literature. Please let us know if you have any other questions. 

Best,

Raymond Adams, Ph.D., MSW
